# Amazon Amandaba—Prevalence, Risk Factors and Self-Care Perception Associated with Diabetic Peripheral Neuropathy in Patients with Type 2 Diabetes: A Cross-Sectional Study

**DOI:** 10.3390/healthcare11040518

**Published:** 2023-02-09

**Authors:** Aline Lobato de Farias, Amanda Suzane Alves da Silva, Victória Brioso Tavares, Josiel de Souza e Souza, Hilton Pereira da Silva, Maria do Socorro Castelo-Branco de Oliveira Bastos, João Simão de Melo-Neto

**Affiliations:** Institute of Health Sciences, Graduate Program in Environmental Health and Society in the Amazon, Federal University of Pará (UFPA), Belém 66075-110, PA, Brazil

**Keywords:** diabetes mellitus type 2, diabetic neuropathies, self-care

## Abstract

Background: Diabetic peripheral neuropathy (DPN) is one of the most common complications of type 2 diabetes mellitus. There is a gradual loss of protective sensation in the skin and the function of the foot joints, increasing the risk of injury as the disease progresses. The objective of this study was to verify whether socioeconomic factors, health risk factors, and self-care are associated with DPN. Methods: Observational cross-sectional with 228 individuals of ≥30 years in Family Health Strategies in a city in the eastern Amazon, in northern Brazil, using questionnaires containing socioeconomic information, clinical and laboratory parameters, the Summary of Diabetes Self-Care Activities Questionnaire, and the Michigan Neuropathy Screening Instrument. Results: The prevalence of DPN was 66.6%. The presence of neuropathy is associated with male gender, dyslipidemia, and increased microalbuminuria. Logistic regression analysis revealed male subjects’ increased BMI and altered HDL levels were associated with DPN. Conclusions: In men with altered BMI, and dysregulation in biochemical parameters, neuropathy is more prevalent.

## 1. Introduction

Type 2 diabetes mellitus (DM2) is the most common type of diabetes [1]. The high prevalence of DM2 is related to urbanization, epidemiological transition, nutritional transition, adoption of a sedentary lifestyle, population growth, and aging [2]. This disease can generate complications and aggravations, such as diabetic peripheral neuropathy (DPN), that can appear with disease progression if not treated and monitored.

DPN is characterized by peripheral nerve dysfunction that leads to a gradual loss of protective sensation in the lower limbs, especially in the feet, increasing the risk of injury [3]. In Brazil, the occurrence of “diabetic foot” as a result of DPN is frequent owing to poor glycemic control, lack of information, non-adherence to treatment, and economic difficulties, in addition to specific problems related to poor hygiene and inappropriate use of shoes [4]. In the early stages, motor deficits are uncommon, but symptoms such as pain, burning, and tingling are present in a third of patients diagnosed with neuropathy [5].

However, if there are injuries, individuals are more susceptible to infections, necrosis, and even limb amputations, which are responsible for the increase in the number of hospitalizations, generating an impact on health resources [6] with significant damage to quality of life, functionality, activities of daily living, family, and social dynamics [4]. The damage can decrease with the expansion of neuropathy screening and guidelines on foot care practices [7]. The risk factors are diverse but depend on the specifics of each region.

Developing countries concentrate on most cases with a tendency to increase in the coming decades and, consequently, the prospect of higher expenses due to complications and treatment of diabetes. In this context, Brazil is among the five countries with the highest number of people living with diabetes [1]. Few studies on DPN screening [8,9,10,11] and self-care for diabetes control using specific instruments in patients treated in primary healthcare in Brazil have been developed. This is especially true in the northern part of the country, the Amazon region, which has social, economic, and cultural issues. These characteristics can affect the self-care practices of these individuals.

The American Diabetes Association (2022) recommends performing DPN screening annually to identify risk factors for ulcers and amputation [12]. In Brazil, the Ministry of Health established, as a screening technique, the evaluation of tactile sensitivity using Semmes–Weinstein monofilament (10 g), vibratory sensitivity with a 128 Hz tuning fork, and Achilles tendon reflex through percussion [13]. However, DPN screening tools that assess signs and symptoms, as well as physical examination, are necessary to assess the damage to patients’ quality of life and map the progression of changes systematically and quantitatively, which is crucial for adequate therapy [14]. Knowing that the application of instruments for the early diagnosis of DPN can direct intervention strategies for the management of exposure to multiple risk factors, reduce the incidence, and delay the progression of DPN [15], this study emerged as a model for evaluating the implementation of the Family Health Strategy (FHS).

Therefore, this study aimed to verify whether socioeconomic status, health risk factors, and self-care were associated with DPN. In addition, the prevalence of diabetic neuropathy in individuals followed up in primary healthcare facilities was identified. Nevertheless, we analyzed whether these risk factors were associated with neuropathy. Our initial hypothesis was that diabetic neuropathy may be associated with socioeconomic and health risk factors. In addition, self-care may be a determinant of DPN in patients with type 2 diabetes.

## 2. Materials and Methods

### 2.1. Study Design

This was an observational cross-sectional study with descriptive and inferential analyses.

### 2.2. Setting and Period of Study

The study used data from the participants’ baseline assessment in the project “Amazon Amandaba: Culture Circles as a strategy to encourage self-care for DM in Primary Health Care,” carried out in November and December 2021. The study was conducted in November and December 2021 in the Family Health Strategy units of the two most populous administrative districts of Belém, Pará in the eastern Amazon. The study abided by the Ethics Committee of the Institute of Health Sciences of the Federal University of Pará (n. 4,693,984).

### 2.3. Population

The study population included individuals with type 2 diabetes mellitus aged 30 years or older (*n* = 2099).

### 2.4. Sampling

We used the simple random probability sampling method. In case of refusal, a new participant was selected by lot.

### 2.5. Sample Size

The sample size was calculated according to Bashar et al. (2021) [16]. The variable “Glycemic control (based on Hb1Ac)” was used as a reference. The values were: odds ratio (OR) = 0.28, proportion p2 = 0.55, error probability α = 0.05, β = 0.20, and allocation ratio N2/N1 = 1.38. The minimum sample size defined was 90 participants.

### 2.6. Sample

Initially, 2099 individuals were evaluated for eligibility and randomization. The initial sample consisted of 230 individuals; 228 participants were selected for analysis and grouped into individuals with (n = 76) or without (n = 152) diabetic neuropathy (Figure 1).

### 2.7. Eligibility Criteria

Individuals aged ≥ 30 years, enrolled in the “Hipertensão e Diabetes” Program (HIPERDIA) assisted in the FHS units, who received the medical diagnosis of type 2 diabetes mellitus at least one year ago, were residents in the two most populous administrative districts, and agreed to participate, were selected for this study by lot. Exclusion criteria were: presenting the neurological diseases, including Alzheimer’s disease, spinal cord injury, stroke, and multiple sclerosis; individuals with complete or significant hearing loss that would interfere with communication during the study; and individuals who were bedridden.

### 2.8. Data Collection and Variables

Social data (age, sex, race/color, education (years), marital status, and income), health habits, clinical characteristics (DM2 diagnosis time (years)), perception of general health status, smoking, systolic arterial hypertension (SAH), and body mass index (BMI, kg/m^2^). In addition, biochemical parameters, including blood glucose (normal: 55–130; high: ≥130 mg/dL), triglycerides (normal: 36–149; high: ≥150 mg/dL), total cholesterol (normal: 87–189; high: ≥190 mg/dL), high-density lipoprotein—HDL (normal: 20–39; high: ≥40 mg/dL), low-density lipoprotein—LDL (normal: 15–99; ≥100 mg/dL), non high-density lipoprotein—NHDL (normal: 59–129; high: ≥130 mg/dL), HbA1c (normal: 4.8–6.9; high ≥7%), and microalbuminuria (normal: <30; micro: 30–300; macro: >300 mg/g), were obtained through laboratory tests.

To measure the perception of self-care, we used the Summary of Diabetes Self-Care Activities Questionnaire, which was translated and adapted to Portuguese [17]. This questionnaire is an instrument that identifies the performance and frequency of activities performed by people with diabetes related to their self-care in the seven days before the evaluation. The answers ranged from 0 to 7, and the scores indicated the performance of these self-care activities. The questionnaire assesses five aspects of the diabetes treatment regimen and groups them into six dimensions of 15 items (general diet, specific diet, exercise, blood glucose testing, foot care, and medication) [17].

#### Michigan Neuropathy Screening Instrument (MSNI)

The Michigan Neuropathy Screening Instrument (MNSI) is a low-cost, quick-to-use tool for evaluating, classifying, and diagnosing diabetic neuropathy. The instrument had two sections (A and B). The first section refers to the participant’s clinical history and contains 15 questions whose answers are summed to obtain the final score for this section. The second section refers to physical examination, starting with the inspection of the feet to observe skin dryness, callus formation, fissures, and the presence of ulcers or deformities. We then performed tests of vibration sensitivity of the hallux with a tuning fork of 128 Hz, in addition to a neurological ankle reflex, and a sensitivity test with a 10 g monofilament [18]. The total scores in Sections A and B were calculated. We assumed a score of 3 (total of 12 in section A) and 2 (total of 10 in section B) as the cutoffs. MSNI values ≤ 5 = no neuropathy, 5.5–10 = moderate neuropathy, and 16.5–22 = severe neuropathy [19].

### 2.9. Study Outcomes

The primary outcome was the presence or absence of diabetic foot neuropathy in patients with DM2.

### 2.10. Bias

Participants were recruited by direct contact after drawing, and if they did not agree to participate in the study, they were replaced by another randomly-selected individual. To minimize possible late-response biases, the benefit of returning the laboratory test results was offered to the research participants and to the health unit to which they were linked to encourage participation in the study.

### 2.11. Statistical Analysis

Descriptive statistical analysis was used to identify the frequency (absolute and relative), mean, standard deviation, and median with interquartile range. Categorical variables were analyzed using the chi-square test (χ^2^), and continuous variables were analyzed using the Mann–Whitney test (U). Binary logistic regression was used to identify independent variables that could be associated with diabetic peripheral neuropathy. The reference value adopted for entering these variables into the multivariate logistic regression model was *p* ≤ 0.25 [20], with no multicollinearity (tolerance > 0.10, variance inflation factor—VIF < 10). A p-value < 0.05 was considered statistically significant for the association between variables in this analysis.

## 3. Results

### 3.1. Prevalence of Diabetic Peripheral Neuropathy

The prevalence of diabetic neuropathy was 152 (66.6%), as determined by the combined score of the two parts of the MSNI, history of symptoms, and physical assessment of the feet.

### 3.2. Social Characteristics

In total, 228 participants were studied. The participant characteristics were as follows: female (52.6%), brown-skin (60.8%), > 61 years of age (57.4%), less than four years of schooling (35.1%), and married (48.2%). The monthly income survey participants lived on was less than one minimum wage (396.46 reais per family member) (Table 1).

The differences between the groups with or without neuropathy showed that men were more frequently affected by neuropathy (*p* = 0.024). Other variables were not statistically significant.

### 3.3. Health-Related and Clinical Characteristics

The timing of the DM2 diagnosis was 10–19 years for the entire sample. Most of the participants were individuals with a smoking history (51.8%) and SAH (69.2%). Regarding participants’ self-perception of their general health, 47.3% considered their health to be regular. The participants had a mean BMI value of 28.87 kg/m^2^, indicating overweight (Table 1). There was no difference between these parameters in participants with and without neuropathy.

### 3.4. Biochemical Parameters

Based on established biochemical parameters, blood glucose (67.9%), triglycerides (62.2%), total cholesterol (51.3%), HDL (51.7%), LDL (62.7%), NHDL (64.9%), and HbA1c (82.8%) levels showed changes in the studied samples. Regarding the microalbuminuria parameters, 71.1% of the participants were within the adopted target values.

When comparing the parameters between individuals with and without neuropathy, altered triglyceride (Zcrit > 1.96), HDL (*p* = 0.031), and NHDL (*p* = 0.049) levels were more prevalent in individuals without neuropathy. However, altered microalbuminuria (*p* = 0.033) was associated with neuropathy.

### 3.5. Diabetes Self-Care Activities

The participants followed self-care practice 2.5 and 2.6 days a week for general and specific diet domains, respectively. The exercise and blood glucose testing domains were the least performed by the participants (0 days), while the foot care and medication domains were the practices most carried out by the participants during a week (4.0 and 4.6 days, respectively) (Table 1).

Participants’ perception of diabetes self-care did not show a statistically significant difference between the groups with or without neuropathy.

### 3.6. Associated Factors with Diabetic Peripheral Neuropathy

Variables that were significant in the univariate binary logistic regression analysis (*p* < 0.25) without multicollinearity (VIF > 10) were entered into the multivariate binary logistic regression model as independent variables, and the outcome (presence or absence of neuropathy) as a dependent variable. Thus, of the 11 variables selected, three were considered to be statistically significant associated with the prevalence of diabetic neuropathy (Table 2). The associated variables were sex, BMI, and HDL cholesterol levels. Male diabetic participants were less likely to develop neuropathy than female participants (OR, 0.50; 95% confidence interval [CI]: 0.26, 0.96, *p* = 0.03). Participants with high BMI were more likely to develop diabetic neuropathy (OR 1.05; 95% CI: 1.00, 1.10, 0.96, *p* = 0.05), and those with established non-standard HDL were more likely to develop neuropathy (OR 2.06; 95% CI: 1.07, 3.97, *p* = 0.03).

## 4. Discussion

This study sought to identify the prevalence of DPN and its relationship with socioeconomic determinants, health habits, biochemical parameters, and perception of self-care in 228 individuals with DM2, who were randomly selected in two administrative districts of a municipality in the northern region of Brazil.

The prevalence of diabetic peripheral neuropathy was 66.6%, with a significant difference according to sex. Men had a higher incidence of DPN (52.6%) compared to women (47.3%). This result was above the average prevalence reported in other studies (between 38% and 42%) in Jordan, Ethiopia, and India [6,16]. Studies in Latin American countries [21,22,23] have shown a prevalence of > 69%. This prevalence may be influenced by the characteristics of each location/region.

The participants, regardless of neuropathy, had a healthy diet, blood glucose monitoring, and less frequent physical activity. These findings are consistent with those of studies conducted in other regions of Brazil [24,25,26]. The use of drugs was a constant practice, which may be related to free access offered by the Unified Health System [27]. Foot care was also a common practice, differing from the average number of days reported in other studies [24,25,26,28].

Our data revealed a high adherence to medication and foot care. Other self-care practices were less frequent. This relationship between poor self-care practices in diabetes and neuropathy requires monitoring, even with high foot care.

High levels of triglycerides, HDL, NHDL, and microalbuminuria were frequent among patients with diabetic neuropathy. In agreement with other reports, it is possible that dyslipidemia is associated with the risk of developing DPN. Evidence supports that treatments for lowering cholesterol and triglycerides help reduce the progression of diabetes complications such as neuropathy [29,30,31]. The controlled levels of microalbuminuria in participants showed a significant association with the presence of neuropathy. Previous studies have demonstrated that DPN is associated with high levels of microalbuminuria [32,33].

Evidence suggests that treatments for lowering cholesterol and triglycerides help reduce the progression of diabetes complications, such as neuropathy [29,30,31]. The controlled levels of microalbuminuria in participants showed a significant association with the presence of neuropathy. Previous studies have demonstrated that DPN is associated with high levels of microalbuminuria [32,33].

The variables Identified as associated with neuropathy were male sex, high HDL levels, and high BMI. Hypercholesterolemia and obesity are significant risk factors for early diabetic neuropathy, independent of glycemic control [6,34,35,36]. HDL contains all atherogenic lipoproteins, which are important parameters for cardiovascular risk. Vascular risk factors are also risk factors for DPN in diabetic and non-diabetic populations [34]. The role of obesity in DPN seems to be related to the activation of inflammatory signaling pathways, given that the chronic inflammatory process seems to be important in the pathogenesis of this complication [37]. Obese patients with diabetes have higher levels of oxidative stress, which is also associated with DPN [38,39]. In relation to males, the differences in lifestyle and testosterone deficiency, which is common in men with diabetes [40], must be taken into account.

Type 2 DM has been associated with long-term macrovascular (atherosclerosis, coronary heart disease, cardiomyopathy, cerebrovascular disease, peripheral artery disease, and lower extremity amputations) and microvascular (retinopathy, nephropathy and neuropathy) complications. This dysfunction is related to an increase in circulating inflammatory factors (C-reactive protein, chemokines, and cytokines), and when associated with obesity-related insulin resistance it is related to worse outcomes [41]. In this study, we observed altered BMI in individuals with NPD. Therefore, the involvement of inflammation should be considered in future research, aiming to verify the role of these markers in the development of this dysfunction.

The limitations of this study are a cross-sectional design, and the sample size of the study is related to the population studied, which should not generalize the findings. Some variables associated with DPN have been identified; however, other structural and socioeconomic issues need to be considered. This affects the patterns that contribute to these risk factors, such as food choice, food insecurity, access to healthy food options, cultural factors, and the social determinants of health.

The use of screening instruments for DPN is low-cost and easily accessible, which prevents late diagnosis and poor prognosis. Knowing the associated aspects of DPN is necessary for developing well-targeted and individualized preventive strategies.

## 5. Conclusions

The prevalence of diabetic neuropathy in these individuals following primary healthcare was 66.6%. Male sex, biochemical parameters, such as triglyceride, HDL, NHDL, and microalbuminuria were associated with DPN. The variables associated with neuropathy were male sex, high BMI, and HDL. Early detection with screening tests and laboratory tests among people with DM is necessary for adequate follow-up.

## Figures and Tables

**Figure 1 healthcare-11-00518-f001:**
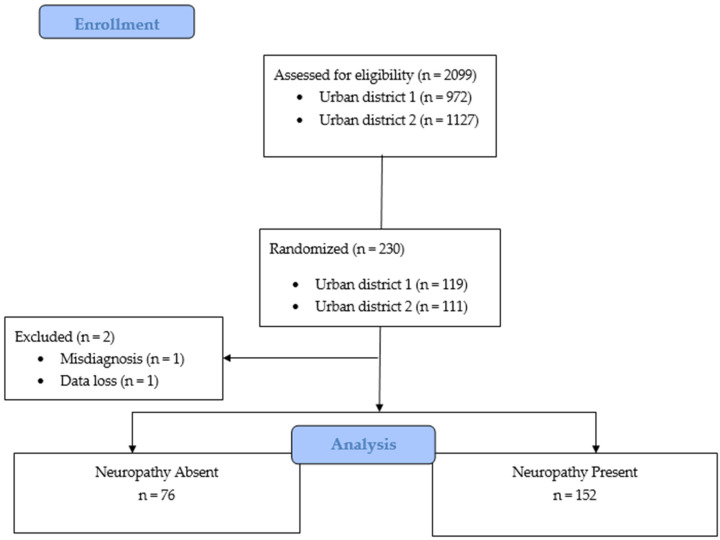
Flow diagram of the selection and distribution of individuals in the groups.

**Table 1 healthcare-11-00518-t001:** Socioeconomic and clinical characteristics and health habits, biochemical parameters of the participants and perception of self-care in diabetes.

Characteristics	Total n (%)	Neuropathy Absent N = 76 ^1^ (%)	Neuropathy Present N = 152 ^1^ (%)	χ^2^	U	*p*-Value
SOCIOECONOMIC CHARACTERISTICS						
Sex				5.06		0.024 *
Male Female	108 (47.3) 120 (52.6)	28 (36.8) 48 (63.1)	80 (52.6) 72 (47.3)			
Race				4.46		0.346
Black Brown White Yellow Indigenous	60 (26.3) 137 (60.8) 26 (11.4) 4 (1.7) 1 (0.4)	23 (30.2) 47 (61.8) 6 (7.8) 0 (0) 0 (0)	37 (24.3) 90 (59.2) 20 (13.1) 4 (2.6) 1 (0.6)			
Age				0.20		0.977
30–40 41–50 51–60 ≥61	6 (2.6) 27 (11.8) 64 (28) 131 (57.4)	2 (2.6) 8 (10.5) 22 (28.9) 44 (57.8)	4 (2.6) 19 (12.5) 42 (27.6) 87 (57.2)			
Education				4.82		0.185
≤4 5–9 10–12 ≥13	80 (35) 75 (32.8) 66 (28.9) 7 (3)	22 (28.9) 25 (32.8) 28 (36.8) 1 (1.3)	58 (38.1) 50 (32.8) 38 (25) 6 (3.9)			
Marital status				3.12		0.680
Married Stable Union Single Widower Divorced Separated	110 (48.2) 47 (20.6) 32 (14) 16 (7) 3 (1.3) 20 (8.7)	34 (44.7) 15 (19.7) 13 (17.1) 6 (7.8) 0 (0) 8 (10.5)	76 (50) 32 (21) 19 (12.5) 10 (6.5) 3 (1.9) 12 (7.8)			
Income	396.4 (275; 550.1)	366.6 (275; 641.8)	406.2 (275; 550)		5531.000	0.601
CLINICAL AND HEALTH HABITS CHARACTERISTICS						
DM2 diagnosis time [years]				6.85		0.076
≤4 5–9 10–19 ≥13	64 (28) 58 (25.4) 70 (30.7) 36 (15.7)	29 (38.1) 19 (25) 20 (26.3) 8 (10.5)	8 (5.2) 39 (25.6) 50 (32.8) 28 (18.4)			
Perception of general health status				0.08		0.960
Good, very good Regular Poor, fair	89 (39) 108 (47.3) 31 (13.5)	29 (38.1) 37 (48.6) 10 (13.1)	60 (39.4) 71 (46.7) 21 (17.7)			
Smoking				1.02		0.599
Non-smoking Ex-smoker Smoker	110 (48.2) 105 (46) 13 (5.7)	36 (47.3) 34 (44.7) 6 (7.8)	74 (30.9) 71 (46.7) 7 (4.6)			
SAH				0.50		0.477
No Yes	70 (30.7) 158 (69.2)	21 (35.5) 55 (72.3)	49 (32.2) 103 (67.7)			
BMI	28.8 (25.7; 32.9)	28 (25.1; 31.5)	29.3 (26; 33.7)		4948.000	0.077
BIOCHEMICAL PARAMETERS						
Blood glucose				0.25		0.615
55–130 mg/dL ≥130 mg/dL	73 (32) 155 (67.9)	26 (34.2) 50 (65.7)	47 (30.9) 105 (69)			
Triglycerides				3.73		0.053*
36–149 mg/dL ≥150 mg/dL	86 (37.7) 142 (62.2)	22 (28.9) 54 (71) ^⧫^	64 (42.1) ^⧫^ 88 (57.8)			
Total cholesterol				1.97		0.159
87–189 mg/dL ≥190 mg/dL	111 (48.6) 117 (51.3)	32 (42.1) 44 (57.8)	79 (51.9) 73 (48)			
HDL				4.64		0.031 *
20–39 mg/dL ≥40 mg/dL	110 (48.2) 118 (51.7)	29 (38.1) 47 (61.8)	81 (53.2) 71 (46.7)			
LDL				0.45		0.497
15–99 mg/dL ≥100 mg/dL	85 (37.2) 143 (62.7)	26 (34.2) 50 (65.7)	59 (38.8) 93 (61.1)			
Non-HDL				3.85		0.049 *
59–129 mg/dL ≥130 mg/dL	80 (35) 148 (64.9)	20 (26.3) 56 (73.6)	60 (39.4) 92 (60.5)			
HbA1c				0.13		0.709
4.8–6.9% ≥7%	39 (17.1) 189 (82.8)	14 (18.4) 62 (81.5)	25 (16.4) 127 (83.5)			
Microalbuminuria				6.79		0.033 *
<30 mg/g 30–300 mg/g >300 mg/g	162 (71) 46 (20.1) 20 (8.7)	61 (80.2) 13 (17.1) 2 (2.6)	101 (66.4) 33 (21.7) 18 (11.8)			
PERCEPTION OF SELF-CARE IN DIABETES						
General Diet	2.5 (0; 4.8)	3 (0; 5)	0.8 (0; 4.5)		5711.500	0.888
Specific diet	2.6 (1.3; 3.3)	2.5 (1.4; 3.3)	2.6 (1.3; 3.6)		5504.000	0.561
Exercise	0 (0; 3.5)	0 (0; 3)	0 (0; 3.5)		5140.500	0.139
Blood Glucose testing	0 (0; 1)	0 (0; 1)	0 (0; 1)		5623.000	0.699
Foot Care	4 (2.3; 5)	4 (2.3; 5)	4 (2.3; 4.9)		5642.500	0.774
Medication	4.6 (4; 4.6)	4.66 (3.4; 4.6)	4.6 (4; 4.6)		5346.000	0.315

^1^ Median (IQR); n (%), * *p*-value ≤ 0.05, ⧫ Adjusted residual post-hoc tests with Z_crit_ = 1.96., X^2^: Qui-square test, U: Mann-Whitney (U) test.

**Table 2 healthcare-11-00518-t002:** Univariate and multivariate binary regression analysis of variables (socioeconomic, clinical, health, and self-care habits) associated with diabetic neuropathy.

	Univariate Analysis (n = 228)			Multivariate Analysis (n = 228)		
**Variables**	**OR ^1^**	**95% CI**	***p*-value**	**OR ^2^**	**95% CI**	***p*-value**
Sex—reference male	1.90	1.08, 3.34	0.025 *	0.50	0.26, 0.96	0.039 **
Education—reference ≥13			0.199 *			0.356
≤4 5–9 10–12	0.43 0.33 0.22	0.05, 3.86 0.03, 2.92 0.02, 1.98	0.458 0.321 0.179 *	0.46 0.34 0.25	0.35, 1.61 0.25, 1.16 0.21, 21.82	0.469 0.118 0.510
DM2 diagnosis time [years]—reference ≥13			0.082 *			0.345
≤4 5–9 10–19	0.34 0.58 0.71	0.13, 0.87 0.22, 1.52 0.27, 1.83	0.024 * 0.275 0.484	0.50 0.79 1.03	0.17, 1.51 0.26, 2.33 0.35, 3.01	0.223 0.671 0.947
BMI	1.03	0.98, 1.07	0.153 *	1.05	1.00, 1.10	0.051 **
Triglycerides	1.78	0.98, 3.22	0.054 *	1.68	0.85, 3.31	0.134
Total cholesterol	1.48	0.85, 2.59	0.160 *	0.81	0.32, 2.05	0.659
HDL	1.84	1.05, 3.24	0.032 *	2.06	1.07, 3.97	0.031 **
Non-HDL	1.82	0.99, 3.34	0.051 *	1.84	0.69, 4.91	0.223
Microalbuminuria—reference >300 mg/g			0.054 *			0.236
<30 mg/g 30–300 mg/g	0.18 0.28	0.04, 0.82 0.05, 1.39	0.026 * 0.120	0.24 0.26	0.05, 1.24 0.04, 1.44	0.090 0.124
Exercise (Summary of Diabetes Self-Care Activities Questionnaire)	1.11	0.97, 1.26	0.118 *	1.12	0.96, 1.30	0.131
Medication (Summary of Diabetes Self-Care Activities Questionnaire)	1.10	0.95, 1.28	0.188 *	1.18	0.99, 1.40	0.061

^1^ OR = Odds Ratio, ^2^ aOR = Adjusted Odds Ratio, CI = Confidence Interval, * *p*-value < 0.25, ** *p*-value ≤ 0.05, 95%CI-Confidence interval.

## Data Availability

Data not available due to ethical principles.

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
