# Peer review of "Amazon Amandaba—Prevalence, Risk Factors and Self-Care Perception Associated with Diabetic Peripheral Neuropathy in Patients with Type 2 Diabetes: A Cross-Sectional Study"

_healthcare, 2023, doi:10.3390/healthcare11040518_

Round 1

Reviewer 1 Report

Authors found that low BMI and ale gender were associated with diabetic neuropathy, a microvascular complication of diabetes mellitus. Authors should discuss the role of inflammation in such an association. Inflamamtory cytokines () and markers () were associated with microvascular complications of type 2 DM. Moreover, obesity is also associated with type 2 DM (). What is the contribution of inflammation in this study? please discuss. 

Moreover, several references are older than 10 years, and if appropriate, they should be either removed or replaced with novel literature. I also suggest consistency in listing the references.

Author Response

Response to Reviewer 1 Comments

Point 1: Authors found that low BMI and ale gender were associated with diabetic neuropathy, a microvascular complication of diabetes mellitus. Authors should discuss the role of inflammation in such an association. Inflamamtory cytokines () and markers () were associated with microvascular complications of type 2 DM. Moreover, obesity is also associated with type 2 DM (). What is the contribution of inflammation in this study? please discuss

Response 1: Thanks for the comment. As required, this information was discussed. Please see the elaborate paragraph highlighted on lines 309 - 317.

Point 2: Several references are older than 10 years, and if appropriate, they should be either removed or replaced with novel literature. I also suggest consistency in listing the references. 

Response 2: Thanks for the comment.As required, references older than 10 years (<2012) have been revised and replaced. However, references 10 (referring to which studies addressing the topic were carried out in Brazil) and 18 (translation and validation of the questionnaire used) were maintained.

Reviewer 2 Report

Lines 22-23:" Logistic regression analysis revealed that male 22 subjects increased BMI and altered HDL levels were predictors of DPN." Are these factors predictors, or associate with DPN?

Line 70: "..were predictors of neuropathy." Kindly ask you to clarify, why you claim that the factors are predictors and not associated with DPN

The above comments are regarding the total manuscript.

Lines 90-95: Kindly ask you to modify, in order to match the text with figure 1.

Line 99:"..who were medically diagnosed with type 2 diabetes mellitus." Kindly ask you to clarify the diagnosis tests.

Line 101: "Participants with neurological disease were excluded.." Please be more specific, because the study is about Neuropathy.

Further, what about patients with vasculopathy? Were they included?

Line 108: "..hypertension (SAH)"

Do the abbreviaton means Systolic Arterial Hypertension? Because it is completly different from Hypertension.

line 116-117: "

To measure the perception of self-care, we used the Summary of Diabetes Self-Care 116
Activities Questionnaire, which was translated and adapted to Portuguese

Line 116-7: "

To measure the perception of self-care, we used the Summary of Diabetes Self-Care 116
Activities Questionnaire, which was translated and adapted to Portuguese

"

As far as my knowledge allows me, the questionnaires in other languages, except from translation need also adjustment. Have you also done that procedure? Kindly ask you for the reference.

Author Response

Response to Reviewer 2 Comments

Point 1: Lines 22-23:" Logistic regression analysis revealed that male 22 subjects increased BMI and altered HDL levels were predictors of DPN." Are these factors predictors, or associate with DPN?. 

Line 70: "..were predictors of neuropathy." Kindly ask you to clarify, why you claim that the factors are predictors and not associated with DPN 

Response 1: Thanks for the comment. As suggested, the term has been changed to associate throughout the entire article for standardization.

Point 2: Lines 90-95: Kindly ask you to modify, in order to match the text with figure 1.

Response 2: Thanks for the comment. We rewrote the text in order to clarify the sample calculation and the sample.

Point 3: Line 99:"..who were medically diagnosed with type 2 diabetes mellitus." Kindly ask you to clarify the diagnosis tests.

Response 3: Thanks for the comment. The individuals received the medical diagnosis for at least 1 year and were enrolled in a public health program aimed at this population. Therefore, we have rewritten this text to make it clearer.

Point 4: Line 101: "Participants with neurological disease were excluded.." Please be more specific, because the study is about Neuropathy.

Response 4: Thanks for the comment. Specific neurological diseases have been described.

Point 5: Further, what about patients with vasculopathy? Were they included?

Response 5: Vasculopathy was not an inclusion or exclusion criterion used for eligibility.

Point 6: Line 108: "..hypertension (SAH)". Do the abbreviaton means Systolic Arterial Hypertension? Because it is completly different from Hypertension.

Response 6: Thanks for the comment. The correct term is Systolic Arterial Hypertension, so we corrected it throughout the manuscript.

Point 7: “To measure the perception of self-care, we used the Summary of Diabetes Self-Care.” “Activities Questionnaire, which was translated and adapted to Portuguese". As far as my knowledge allows me, the questionnaires in other languages, except from translation need also adjustment. Have you also done that procedure? Kindly ask you for the reference.

Response 7: Thanks for the comment. The questionnaire had already been translated and validated into Portuguese, the native language in Brazil. Therefore, there was no need for any adaptation. We quote the reference at the end of this sentence in order to clarify possible doubts.

Round 2

Reviewer 2 Report

Line 180: Abbreviation is not explained before. "SAH"

Response: Thank you for your suggestion. We have added it.